# Identification of high-risk factors for recurrence of colon cancer following complete mesocolic excision: An 8-year retrospective study

**Yuan Liu**[1☯], **Wenyi Du**[1☯], **Yi Guo**[2], **Zhiqiang Tian**[1], **Wei Shen**[1]*

1 Department of General Surgery, Wuxi People's Hospital Affiliated to Nanjing Medical University, Wuxi, China, 2 Department of General Practice, Shandong Provincial Hospital Affiliated to Shandong First Medical University, Jinan, Shandong Province, China

☯ These authors contributed equally to this work.
* shenweiijs@outlook.com

**Data Availability Statement:** All relevant data are within the paper and its Supporting Information files.

## Abstract

### Background

Colon cancer recurrence is a common adverse outcome for patients after complete mesocolic excision (CME) and greatly affects the near-term and long-term prognosis of patients. This study aimed to develop a machine learning model that can identify high-risk factors before, during, and after surgery, and predict the occurrence of postoperative colon cancer recurrence.

### Methods

The study included 1187 patients with colon cancer, including 110 patients who had recurrent colon cancer. The researchers collected 44 characteristic variables, including patient demographic characteristics, basic medical history, preoperative examination information, type of surgery, and intraoperative information. Four machine learning algorithms, namely extreme gradient boosting (XGBoost), random forest (RF), support vector machine (SVM), and k-nearest neighbor algorithm (KNN), were used to construct the model. The researchers evaluated the model using the k-fold cross-validation method, ROC curve, calibration curve, decision curve analysis (DCA), and external validation.

### Results

Among the four prediction models, the XGBoost algorithm performed the best. The ROC curve results showed that the AUC value of XGBoost was 0.962 in the training set and 0.952 in the validation set, indicating high prediction accuracy. The XGBoost model was stable during internal validation using the k-fold cross-validation method. The calibration curve demonstrated high predictive ability of the XGBoost model. The DCA curve showed that patients who received interventional treatment had a higher benefit rate under the XGBoost model. The external validation set's AUC value was 0.91, indicating good extrapolation of the XGBoost prediction model.

**Funding:** This work was supported by the Top Talent Support Program for young and middle-aged people of Wuxi Health Committee (Grant No. HB2020007).The funders had no role in study design, data collection and analysis, decision to publish, or preparation of the manuscript.

**Competing interests:** The authors have declared that no competing interests exist.

## Conclusion

The XGBoost machine learning algorithm-based prediction model for colon cancer recurrence has high prediction accuracy and clinical utility.

## Introduction

Colon cancer is a gastrointestinal tumor that carries a grave prognosis. The incidence of colorectal cancer is on the rise due to changes in lifestyle and dietary habits, and there is a gradual shift in the incidence from the distal rectum to the proximal colon. According to the 2019 epidemiological survey [1], colon cancer ranks as the third most common malignancy worldwide, after lung cancer and breast cancer. To decrease the morbidity and mortality of colon cancer patients, Hohenberger proposed complete mesocolic excision (CME), which involves the removal of the tumor along with the colonic mesentery, followed by the ligation of tumor vessels at the root to ensure radical surgery. As surgical techniques continue to develop, open surgery has given way to laparoscopic and robot-assisted surgery, leading to further improvements in the prognosis of colon cancer patients [2, 3]. However, despite the effectiveness of radical colon cancer surgery, clinicians have discovered that some patients have poor outcomes, such as tumor recurrence and distant metastases, which have extremely high mortality rates [4]. According to one study [5], tumor recurrence is the primary cause of postoperative death in colon cancer patients. Thus, it is essential to identify the risk factors for colon cancer recurrence and predict its occurrence.

Artificial intelligence (AI) is advancing rapidly in the medical field [6]. As a significant branch of AI, machine learning offers more stable model building and more accurate prediction, making it a popular choice among clinicians and widely used in clinical prediction and other areas [7, 8]. In this study, we analyzed the clinical data of colon cancer patients and employed machine learning algorithms to develop a prediction model for colon cancer recurrence. This will enable clinicians to formulate precise individualized treatment plans promptly and improve the postoperative survival rate of patients.

## Materials and methods

### Study subjects

In the current study, we utilized clinical data from a database of colon cancer patients at Wuxi People's Hospital from January 2010 to January 2018. The inclusion criteria for cases were as follows: (1) patients who underwent open CME or laparoscopic-assisted CME; (2) the surgical team consisted of senior doctors who were able to independently perform CME; and (3) patients were diagnosed with colon cancer by imaging and tumor pathology. The exclusion criteria for cases were as follows: (1) patients with other malignant tumors; (2) patients with serious cardiovascular and cerebrovascular diseases or liver, kidney, and other significant organ diseases; and (3) case records with missing or lost visits. The patients in this study were monitored for a minimum of 5 years after undergoing surgery, during which time they were regularly examined by two surgeons who conducted medical history reviews, physical examinations, and imaging tests including abdominal ultrasounds and computed tomography (CT) scans every three months. The Ethics Committee of Wuxi People's Hospital approved this study, with approval number KY22085. As this retrospective investigation was conducted, and in adherence to local laws and regulations, the Ethics Committee granted a waiver for the necessity of informed consent, as we have diligently anonymized all patient data.

## Study design and data collection

A total of 44 preoperative variables (within 24 h of the day of surgery), intraoperative variables, and postoperative variables (within 48 h of the initial surgery) were collected. Preoperative variables collected included patient demographics (gender, age, smoking history, alcohol history, and body mass index), basic clinical characteristics (American Society of Anesthesiologists score, nutrition risk screening 2002 score, surgical history, disease duration, adjuvant chemotherapy history, and adjuvant radiotherapy history), basic medical history (anemia, diabetes, ileus, hypertension, hyperlipidemia, and coronary artery disease), laboratory tests (albumin, carcinoembryonic antigen, carbohydrate antigen 19–9, carbohydrate antigen 125, and carbohydrate antigen 72–4), tumor characteristics (T-stage, N-stage, peripheral nerve invasion, vascular invasion, tumor size, tumor number, tumor configuration, and pathologic type). Intraoperative variables collected included surgical approach, type of surgery, duration of surgery, intraoperative bleeding, number of surgically cleared lymph nodes, and whether it was an emergency surgery. Postoperative variables collected included laboratory test indices (carcinoembryonic antigen, carbohydrate antigen 19–9, carbohydrate antigen 125, carbohydrate antigen 72–4, procalcitonin, C-reactive protein, serum amyloid A, and neutrophil to lymphocyte ratio) and tumor characteristics (tumor recurrence).

## Development and evaluation of predictive models for machine learning algorithms

The statistical software programs SPSS and R were utilized to develop and assess the clinical prediction models. (1) Univariate and multivariate regression analyses were conducted. Categorical variables were compared between the two groups using the chi-square test, while the t-test was used for continuous variables that followed a normal distribution. For continuous variables that did not meet the normal distribution criteria, the rank sum test was used. Statistical significance was determined by a p-value of less than 0.05. Logistic regression analysis was performed on variables that showed significance in the univariate analysis to identify independent factors that influenced the occurrence of postoperative colon cancer recurrence. Four predictive models, namely extreme gradient boosting (XGBoost), random forest (RF), support vector machine (SVM), and k-nearest neighbor algorithm (KNN), were utilized to score and rank the significance of all the variables. Variables that appeared in the top ten rankings in all four models and were also significant in both univariate and multivariate regression analyses were chosen. (2) Evaluation and development of prediction models. Colon cancer patients diagnosed between January 2010 and December 2016 were selected as the internal validation set, while patients diagnosed between January 2017 and January 2018 were chosen as the external validation set. The internal validation set was divided randomly into a training set (70%) and a test set (30%). The top ten variables, selected based on their significance in univariate and multivariate regression analyses and ranking in the top ten in all four machine learning algorithm models (SVM, RF, XGBoost, and KNN), were incorporated into the four prediction models. Three aspects were used to evaluate the models: discrimination, calibration, and clinical usefulness. The best model was selected for prediction analysis. Receiver operating characteristic (ROC) curves were plotted to determine the area under the curve (AUC) values and predictive efficacy of the models. Calibration curves were used to assess whether the models predicted actual results with good agreement, while decision curve analysis (DCA) was used to assess the benefits of patients after interventional treatment. Internal validation was completed using the k-fold cross-validation method. (3) External validation of the optimal model was conducted using an independent test set. The ROC curve was plotted to evaluate the predictive accuracy and generalizability of the model. (4) Model interpretation. The Shapley value is utilized in

SHAP analysis to obtain the contribution of each feature in the sample to the prediction. Based on the Shapley values, the SHAP summary plot is generated to rank the importance of risk factors, and the SHAP force plot is constructed to analyze and interpret the prediction results of individual samples.

## Results

### Basic clinical information of the patient

A total of 1187 patients were included in the study, including 110 (9.27%) patients with recurrent colon cancer (Fig 1). The original data presented in the study are included in the S1 Table.

### Analysis of risk factors for postoperative recurrence of colon cancer

The results of both univariate and multivariate analyses indicated that T-stage, N-stage, liver metastases, vascular invasion, tumor number, tumor size, preoperative carcinoembryonic antigen (CEA) level, postoperative CEA level, preoperative carbohydrate antigen 19–9 (CA19-9) level, postoperative CA19-9 level, albumin (ALB), and emergency surgery had significant independent effects on colon cancer recurrence (P<0.05) (Table 1). The XGBoost, RF, SVM, and KNN models were used to identify the risk factors affecting the recurrence of colon cancer, and the top variables selected were N-stage, liver metastases, tumor number, tumor size, postoperative carbohydrate antigen 125 (CA125) level, C-reactive protein (CRP) level, neutrophil to lymphocyte ratio (NLR), and postoperative CEA level (Fig 2A–2D). Based on these results, the variables used to construct the predictive model in this study were N-stage, liver metastases, tumor number, tumor size, and postoperative CEA level.

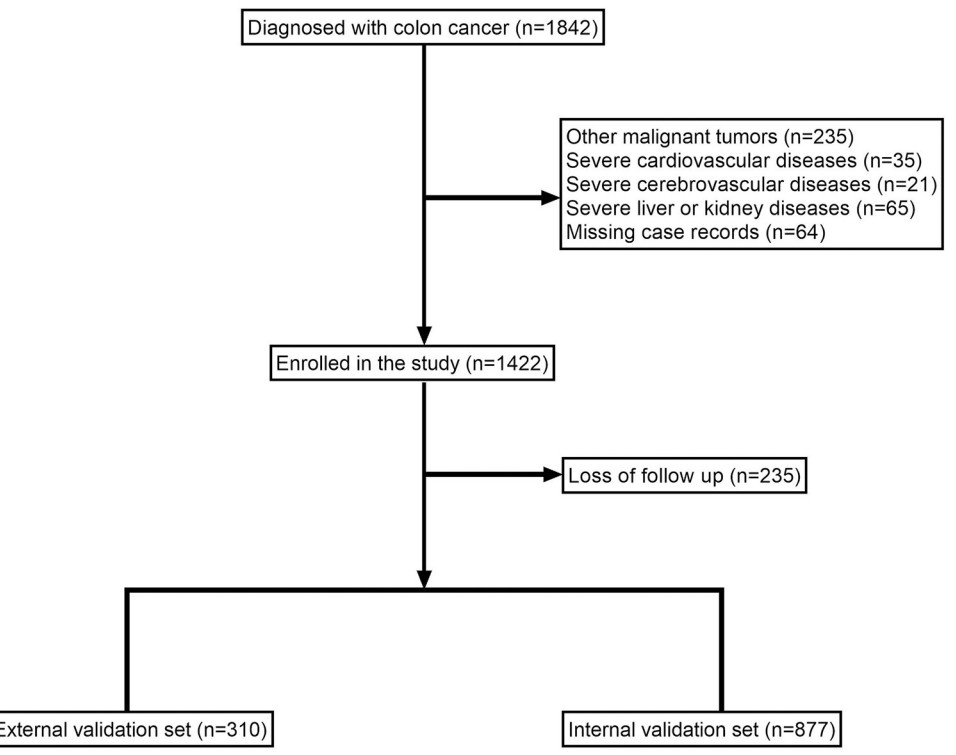

**Fig 1. Flow diagram of patients included in the study.**

**Table 1. Univariate and multivariate analyses of variables related to recurrence of colon cancer.**

| Variables | | Univariate analysis | | | Multivariate analysis | | |
|---|---|---|---|---|---|---|---|
| | | OR | 95%CI | P-value | OR | 95%CI | P-value |
| Sex | Female | Reference | | | | | |
| | Male | 1.253 | [0.793,1.980] | 0.334 | | | |
| Age | <65 | Reference | | | Reference | | |
| | ≥65 | 4.471 | [2.803,7.132] | <0.001 | 0.569 | [0.074,3.375] | 0.554 |
| BMI | <25 kg/m$^2$ | Reference | | | | | |
| | ≥25 kg/m$^2$ | 1.379 | [0.822,2.313] | 0.224 | | | |
| ASA | <3 | Reference | | | | | |
| | ≥3 | 0.933 | [0.592,1.470] | 0.764 | | | |
| Family history | No | Reference | | | | | |
| | Yes | 1.219 | [0.586,2.538] | 0.596 | | | |
| Drinking history | No | Reference | | | | | |
| | Yes | 1.336 | [0.808,2.209] | 0.259 | | | |
| Smoking history | No | Reference | | | | | |
| | Yes | 1.218 | [0.727,2.039] | 0.454 | | | |
| Surgical history | No | Reference | | | | | |
| | Yes | 1.167 | [0.646,2.107] | 0.609 | | | |
| Anemia | No | Reference | | | | | |
| | Yes | 1.203 | [0.684,2.114] | 0.521 | | | |
| Hyperlipidemia | No | Reference | | | | | |
| | Yes | 1.066 | [0.591,1.921] | 0.832 | | | |
| Hypertensive | No | Reference | | | | | |
| | Yes | 0.658 | [0.378,1.148] | 0.141 | | | |
| Diabetes | No | Reference | | | | | |
| | Yes | 1.468 | [0.897,2.403] | 0.127 | | | |
| CHD | No | Reference | | | | | |
| | Yes | 1.487 | [0.755,2.930] | 0.251 | | | |
| T-stage | T1~T2 | Reference | | | Reference | | |
| | T3~T4 | 10.565 | [6.356,17.560] | <0.001 | 13.08 | [2.119,122.006] | 0.011 |
| N-stage | N0 | Reference | | | Reference | | |
| | N1~N2 | 11.794 | [7.199,19.322] | <0.001 | 54.031 | [8.325,638.782] | <0.001 |
| Liver metastasis | No | Reference | | | Reference | | |
| | Yes | 10.019 | [5.979,16.790] | <0.001 | 143.5 | [13.87,3422.934] | <0.001 |
| Vascular invasion | No | Reference | | | Reference | | |
| | Yes | 21.71 | [12.875,36.610] | <0.001 | 41.956 | [6.13,482.76] | 0.001 |
| PNI | No | Reference | | | Reference | | |
| | Yes | 16.033 | [9.279,27.703] | <0.001 | 5.731 | [0.597,63.452] | 0.133 |
| Ileus | No | Reference | | | | | |
| | Yes | 0.825 | [0.513,1.326] | 0.427 | | | |
| Adjuvant Radiotherapy | No | Reference | | | Reference | | |
| | Yes | 3.984 | [2.415,6.572] | <0.001 | 2.493 | [0.358,16.583] | 0.341 |
| Adjuvant Chemotherapy | No | Reference | | | Reference | | |
| | Yes | 6.156 | [3.801,9.968] | <0.001 | 0.751 | [0.156,3.326] | 0.710 |
| Surgical procedure | Laparoscopic surgery | Reference | | | | | |
| | Open surgery | 0.727 | [0.460,1.147] | 0.171 | | | |
| Emergency surgery | No | Reference | | | Reference | | |
| | Yes | 11.881 | [7.181,19.658] | <0.001 | 7.674 | [1.768,40.754] | 0.009 |

*(Continued)*

**Table 1.** (Continued)

| Variables | | Univariate analysis | | | Multivariate analysis | | |
|---|---|---|---|---|---|---|---|
| | | OR | 95%CI | P-value | OR | 95%CI | P-value |
| **Lymph node dissection** | <12 | Reference | | | | | |
| | ≥12 | 0.82 | [0.486,1.384] | 0.457 | | | |
| **Tumor number** | <2 | Reference | | | Reference | | |
| | ≥2 | 6.142 | [3.821,9.872] | <0.001 | 9.042 | [1.656,70.975] | 0.018 |
| **Tumor size** | <5 cm | Reference | | | Reference | | |
| | ≥5 cm | 18.836 | [10.881,32.606] | <0.001 | 14.175 | [2.825,98.71] | 0.003 |
| **Preoperative CEA level** | <5 ng/ml | Reference | | | Reference | | |
| | ≥5 ng/ml | 4.014 | [2.521,6.390] | <0.001 | 8.002 | [1.678,53.106] | 0.016 |
| **Postoperative CEA level** | <5 ng/ml | Reference | | | Reference | | |
| | ≥5 ng/ml | 9.775 | [5.968,16.011] | <0.001 | 11.029 | [2.263,77.939] | 0.006 |
| **Preoperative CA19-9 level** | <37 U/mL | Reference | | | Reference | | |
| | ≥37 U/mL | 9.227 | [5.682,14.984] | <0.001 | 7.815 | [1.773,43.559] | 0.010 |
| **Postoperative CA19-9 level** | <37 U/mL | Reference | | | Reference | | |
| | ≥37 U/mL | 15.123 | [9.137,25.029] | <0.001 | 14.589 | [2.696,115.06] | 0.004 |
| **Preoperative CA125 level** | <35 U/ml | Reference | | | Reference | | |
| | ≥35 U/ml | 4.112 | [2.567,6.587] | <0.001 | 1.562 | [0.291,8.4] | 0.595 |
| **Postoperative CA125 level** | <35 U/ml | Reference | | | Reference | | |
| | ≥35 U/ml | 4.265 | [2.672,6.809] | <0.001 | 0.664 | [0.134,2.812] | 0.590 |
| **Preoperative CA72-4 level** | <7 U/ml | Reference | | | Reference | | |
| | ≥7 U/ml | 7.64 | [4.429,13.179] | <0.001 | 5.396 | [0.849,39.847] | 0.080 |
| **Postoperative CA72-4 level** | <7 U/ml | Reference | | | | | |
| | ≥7 U/ml | 1.421 | [0.622,3.248] | 0.405 | | | |
| **PCT level** | <0.05 ng/ml | Reference | | | | | |
| | ≥0.05 ng/ml | 1.165 | [0.610,2.224] | 0.644 | | | |
| **CRP level** | <10 mg/l | Reference | | | Reference | | |
| | ≥10 mg/l | 2.286 | [1.441,3.625] | <0.001 | 4.454 | [0.735,31.332] | 0.111 |
| **SAA level** | <10 mg/l | Reference | | | Reference | | |
| | ≥10 mg/l | 2.19 | [1.282,3.742] | 0.004 | 8.031 | [0.693,123.636] | 0.101 |
| **NLR** | <3 | Reference | | | Reference | | |
| | ≥3 | 5.863 | [3.602,9.543] | <0.001 | 0.53 | [0.068,3.336] | 0.514 |
| **NRS2002 score** | <3 | Reference | | | | | |
| | ≥3 | 0.751 | [0.468,1.207] | 0.237 | | | |
| **ALB** | ≥30 g/l | Reference | | | Reference | | |
| | <30 g/l | 0.519 | [0.310,0.869] | 0.013 | 0.091 | [0.007,0.776] | 0.043 |
| **Pathologic type** | Adenocarcinoma | Reference | | | | | |
| | Mucinous adenocarcinoma | 1.16 | [0.662,2.033] | 0.604 | | | |
| | Other | 1.133 | [0.643,1.994] | 0.666 | | | |
| **Tumor configuration** | Exophytic | Reference | | | | | |
| | Ulcerative | 1.01 | [0.519,1.964] | 0.977 | | | |
| | Infiltrative | 1.309 | [0.695,2.465] | 0.405 | | | |
| | Unknown | 1.074 | [0.556,2.073] | 0.832 | | | |
| **Disease duration** | <6 months | Reference | | | | | |
| | ≥6 months | 0.913 | [0.579,1.438] | 0.693 | | | |

Abbreviations: OR, odds ratio; CI, confidence interval; BMI, body mass index; ASA, The American Society of Anesthesiologists; ALB, albumin; CA125, carbohydrate antigen 125; CA19-9, carbohydrate antigen 19–9; CA72-4, carbohydrate antigen 72–4; PCT, procalcitonin; CRP, C-reactive protein; SAA, serum amyloid A; NRS2002, nutrition risk screening 2002; CHD, coronary heart disease; CEA, carcinoembryonic antigen; PNI, peripheral nerve invasion; NLR, neutrophil-to-lymphocyte ratio.

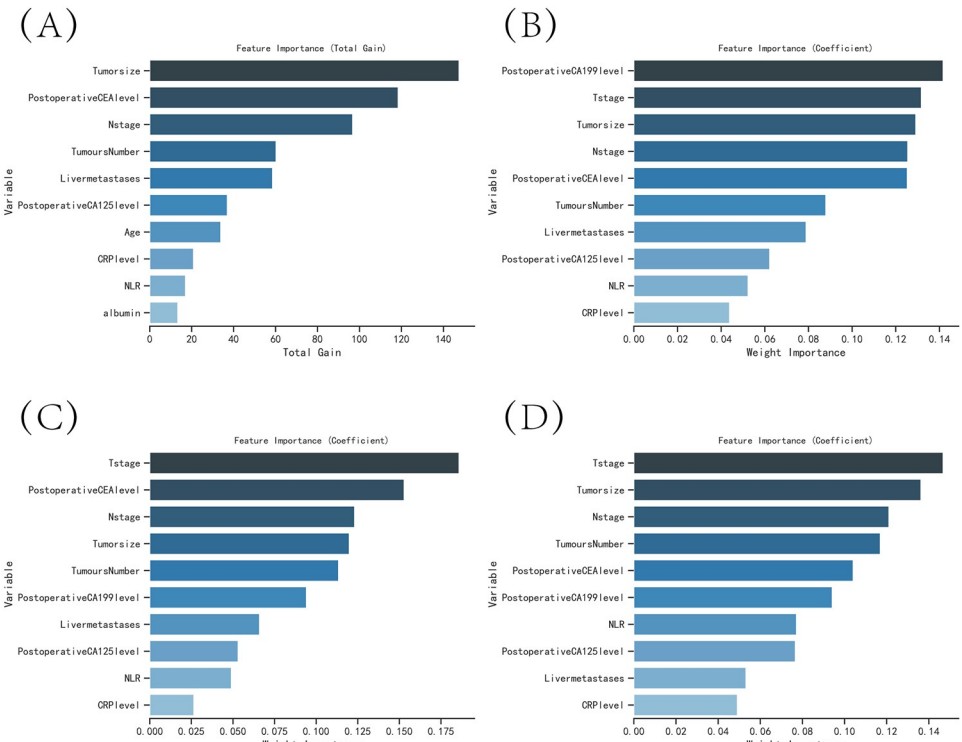

**Fig 2. The variable influence factor ranking plots of the four models.** (A) Variable importance ranking diagram of the XGBoost model. (B) Variable importance ranking diagram of the RF model. (C) Variable importance ranking diagram of the SVM model. (D) Variable importance ranking diagram of the KNN model.

## Model building and evaluation

The results of the ROC curve analysis showed that the XGBoost model had the highest AUC value in both the training set (0.962) and the validation set (0.952), indicating good discrimination ability (Table 2). The calibration curve analysis showed that the predicted probabilities from the XGBoost model were well-calibrated with the actual probabilities. The Brier score of XGBoost was the lowest among the four models, indicating good accuracy of the predicted probabilities. The DCA curves showed that all four models had a net clinical benefit, with

**Table 2. Evaluation of the performance of the four models.**

|  |  | AUC(95%CI) | Accuracy(95%CI) | Sensitivity(95%CI) | Specificity(95%CI) |
|---|---|---|---|---|---|
| **KNN** | training set | 0.897 (0.841–0.954) | 0.943(0.939–0.948) | 0.838(0.803–0.872) | 0.938(0.916–0.960) |
|  | validation set | 0.840 (0.706–0.970) | 0.935(0.924–0.946) | 0.736(0.652–0.821) | 0.932(0.912–0.951) |
| **XGBoost** | training set | 0.962 (0.937–0.987) | 0.929(0.924–0.933) | 0.898(0.885–0.912) | 0.923(0.917–0.929) |
|  | validation set | 0.952 (0.897–0.999) | 0.923(0.907–0.938) | 0.925(0.881–0.969) | 0.922(0.897–0.946) |
| **RF** | training set | 0.941 (0.907–0.976) | 0.918(0.905–0.930) | 0.867(0.844–0.889) | 0.904(0.880–0.928) |
|  | validation set | 0.945 (0.890–0.996) | 0.904(0.881–0.927) | 0.959(0.924–0.994) | 0.870(0.812–0.928) |
| **SVM** | training set | 0.854 (0.776–0.932) | 0.934(0.902–0.967) | 0.766(0.710–0.823) | 0.946(0.904–0.987) |
|  | validation set | 0.802 (0.622–0.970) | 0.911(0.875–0.948) | 0.743(0.658–0.828) | 0.908(0.860–0.956) |

Abbreviations: CI, confidence interval; KNN, k-nearest neighbor; XGBoost, extreme gradient boosting; RF, random forest; SVM, support vector machine; AUC, area under the curve.

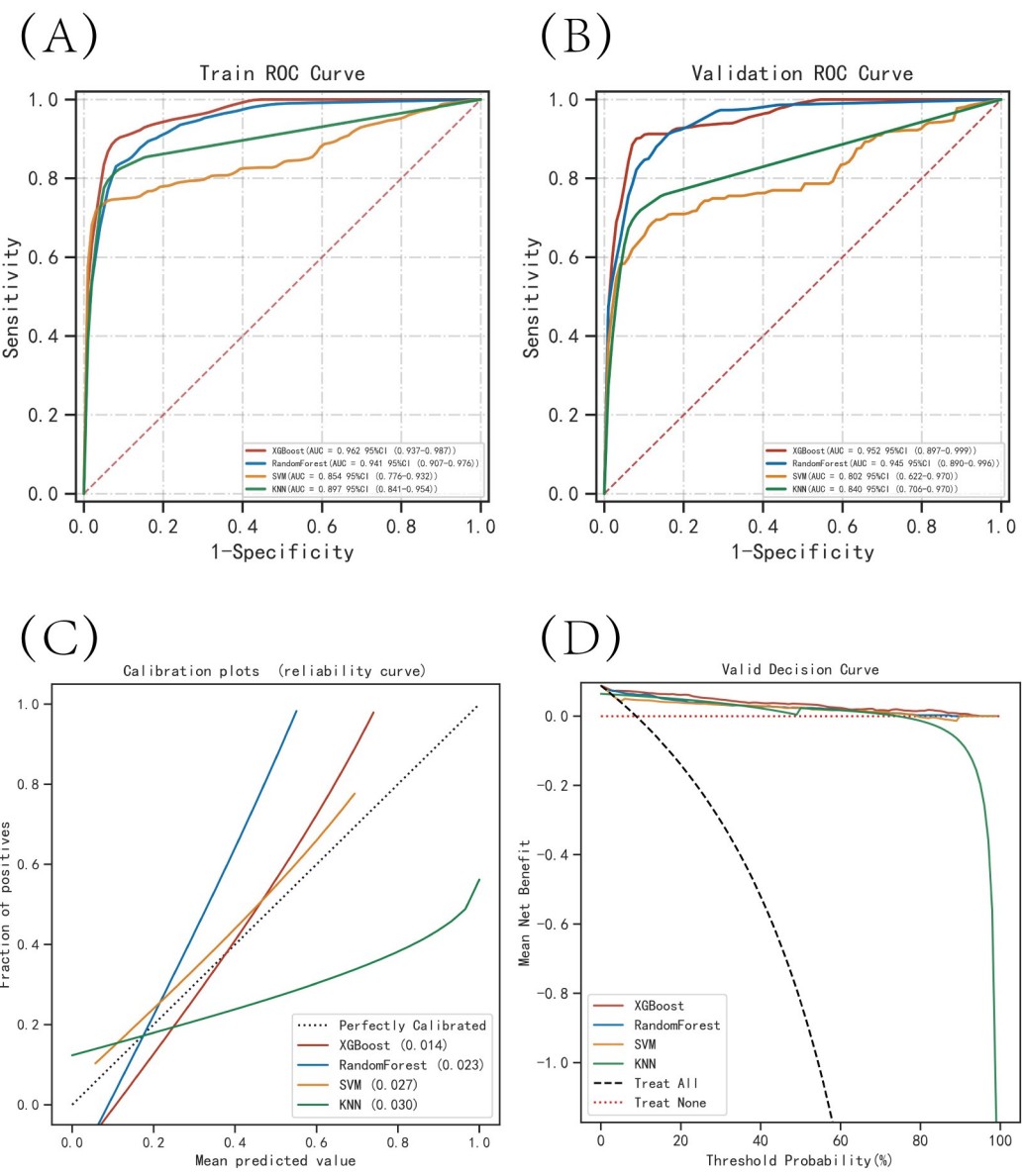

**Fig 3. Evaluation of the four models for predicting recurrence of colon cancer.** (A) ROC curves for the training set of the four models. (B) ROC curves for the validation set of the four models. (C) Calibration plots of the four models. The 45° dotted line on each graph represents the perfect match between the observed (y-axis) and predicted (x-axis) complication probabilities. A closer distance between two curves indicates greater accuracy. (D) DCA curves of the four models. The intersection of the red curve and the All curve is the starting point, and the intersection of the red curve and the None curve is the node within which the corresponding patients can benefit.

XGBoost having the highest net benefit at most probability thresholds (Fig 3A–3D). The k-fold cross-validation method was used to evaluate the generalization ability of the four models. A test set of 264 cases (30.10%) was randomly selected from the overall dataset, and the remaining samples were used as the training set for 10-fold cross-validation. The XGBoost model had an AUC of 0.9358±0.0391 for the validation set and an AUC of 0.9158 for the test set, with an accuracy of 0.8939 (Fig 4A–4C). The RF model had an AUC of 0.9177±0.0709 for the validation set and an AUC of 0.8734 for the test set, with an accuracy of 0.8939. The SVM model had an AUC of 0.8451±0.1078 for the validation set and an AUC of 0.8183 for the test

set, with an accuracy of 0.9583. The KNN model had an AUC of 0.8801±0.0661 for the validation set and an AUC of 0.8715 for the test set, with an accuracy of 0.9242. After comprehensive comparison, the XGBoost algorithm was chosen to construct the predictive model in this study.

## Model external validation

The results obtained from the ROC curve showed an AUC value of 0.91 for the external validation set, which is a strong indication that the prediction model has high accuracy in determining the occurrence of the disease (Fig 4D).

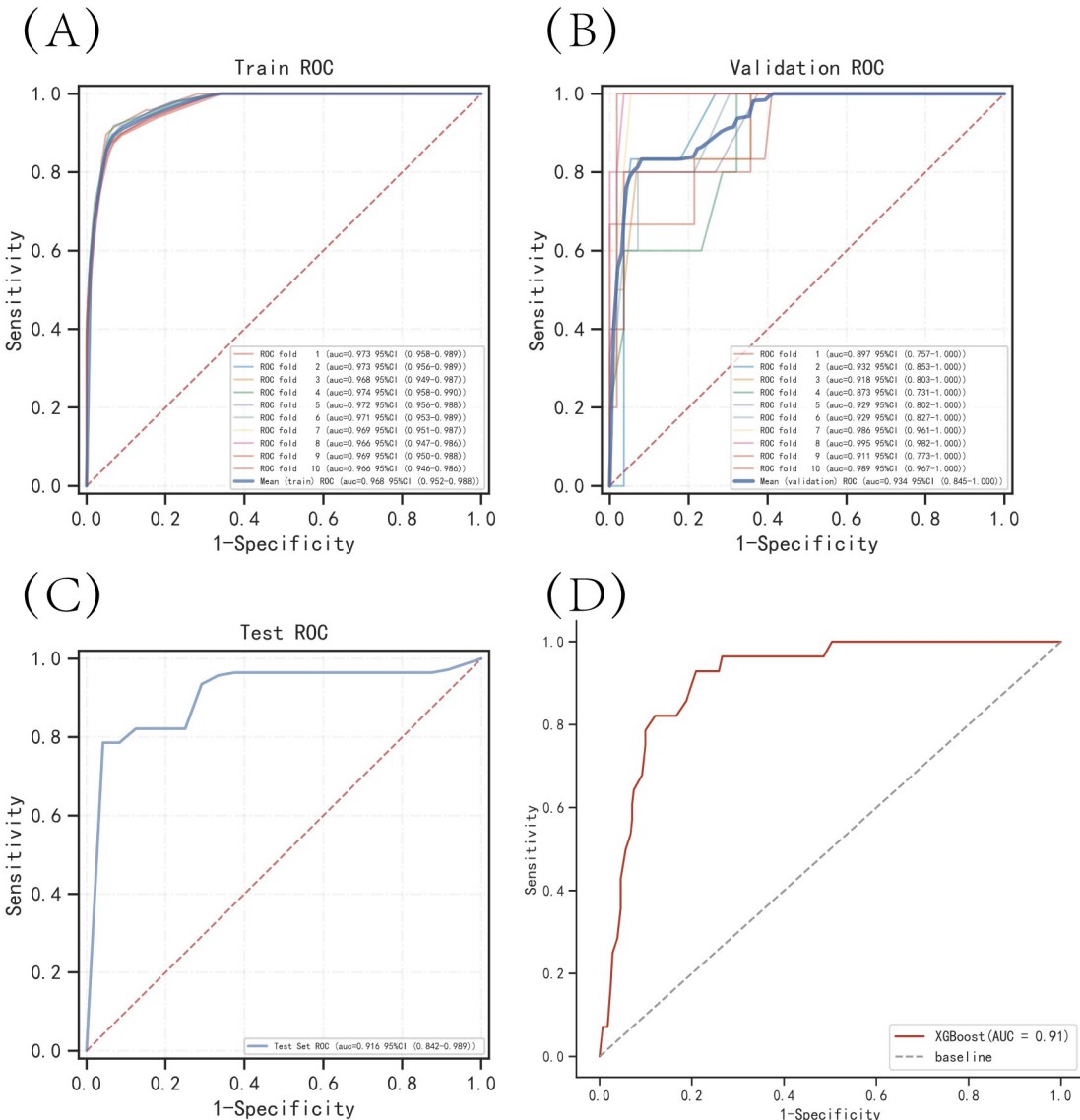

**Fig 4. Internal validation of the XGBoost model.** (A) ROC curve of the XGBoost model for the training set. (B) ROC curve of the XGBoost model for the validation set. (C) ROC curve of the XGBoost model for the test set. (D) External validation of the XGBoost model.

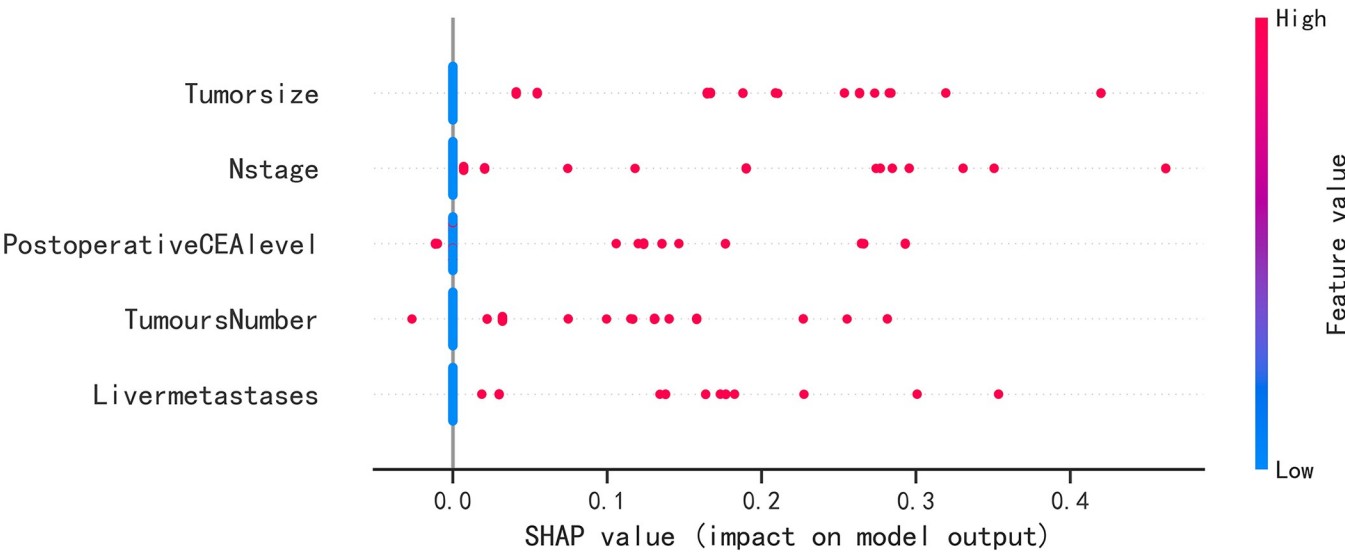

**Fig 5. SHAP summary plot.** Risk factors are arranged along the y-axis based on their importance, which is given by the mean of their absolute Shapley values. The higher the risk factor is positioned in the plot, the more important it is for the model.

## Model explanation

The SHAP summary plot revealed that the risk factors for the recurrence of colon cancer were ranked in the following order: tumor size, N-stage, postoperative CEA level, tumor number, and liver metastases (Fig 5). The SHAP force plots depict the predictive analysis of the study model for four patients who had recurrent colon cancer. For patient I, the model predicted a 0.076 probability of recurrence, with an increased probability of tumor volume $\geq$ 5 cm and tumor lymphatic metastasis. For patient II, the model predicted a 0.007 probability of recurrence, with an increased probability of tumor lymphatic metastasis. For patient III, the model predicted a 0.365 probability of recurrence, with an increased probability of tumor volume $\geq$ 5 cm and tumor liver metastasis. For patient IV, the model predicted a 0.747 probability of recurrence, with an increased probability of tumor volume $\geq$ 5 cm, tumor lymphatic metastasis, and postoperative CEA $\geq$ 5 ng/ml (Fig 6A–6D).

## Discussion

This study aimed to evaluate the risk prediction models constructed by four machine learning algorithms, and among them, the XGBoost algorithm was found to exhibit exceptional accuracy and efficiency. Unlike the RF algorithm, the XGBoost algorithm takes into account the regularization problem and effectively avoids overfitting of the model [9]. In comparison to the SVM algorithm and KNN algorithm, the XGBoost algorithm is better suited for large sample sizes and multiple feature variables, which reduces the computational and training time required [10]. Therefore, the XGBoost algorithm was chosen to construct a model to predict the recurrence of colon cancer after surgery. The prediction model serves at least two purposes, one of which is to clarify the risk factors for tumor recurrence, and the other is to prompt clinicians to take timely interventions for high-risk patients to reduce the risk of tumor recurrence. In this study, SHAP analysis was used to interpret the model, and the results showed that CEA $\geq$5 ng/ml, tumor size, lymphatic metastasis, liver metastasis, and multiple tumors were identified as risk factors for the recurrence of colon cancer after radical colon cancer surgery.

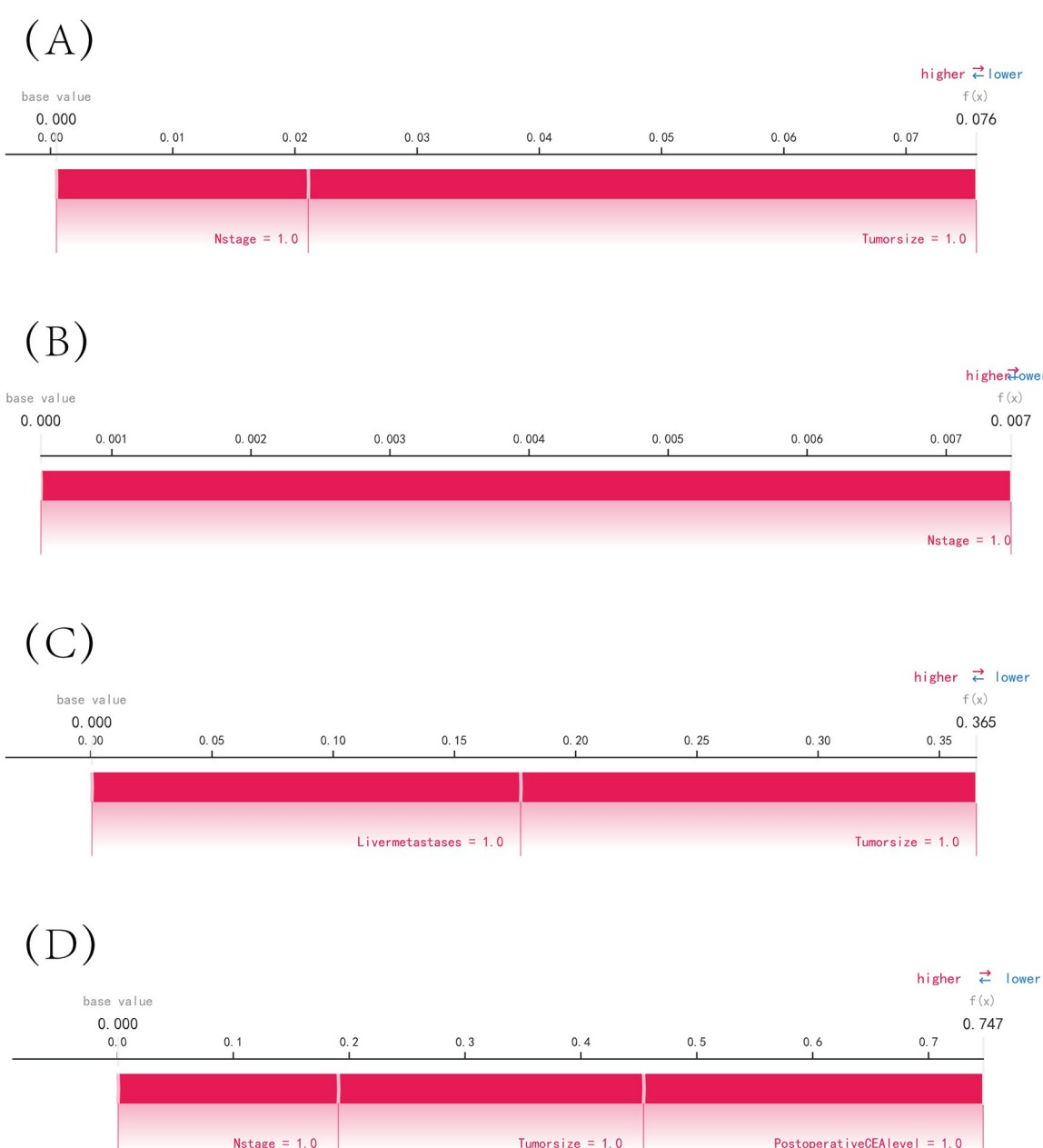

**Fig 6. SHAP force plot.** The contributing variables are arranged in the horizontal line, sorted by the absolute value of their impact. Blue represents features that have a negative effect on disease prediction, with a decrease in SHAP values; red represents features that have a positive effect on disease prediction, with an increase in SHAP values. (A) Predictive Analysis of Patient I. (B) Predictive Analysis of Patient II. (C) Predictive Analysis of Patient III. (D) Predictive Analysis of Patient IV.

The greater the size of a tumor, the deeper it infiltrates the surrounding tissues, thereby increasing the probability of lymphatic and distant metastasis, and rendering complete surgical intervention more difficult. The National Comprehensive Cancer Network (NCCN) and the American Joint Committee on Cancer (AJCC) have laid out explicit guidelines regarding the radicality of colon cancer surgery, emphasizing that the procedure should excise a sufficiently extensive section of bowel to ensure negative surgical margins [11, 12]. However, the depth of tumor invasion may be too extensive to enable the surgeon to precisely determine the extent of

the lesion resection with the naked eye. Moreover, performing intraoperative rapid pathological examination to guarantee negative margins is often challenging, resulting in an augmented risk of postoperative tumor remnants. Additionally, larger tumors tend to divide at a quicker pace, generating more tumor vessels. Tomisaki's analysis of 175 colon cancer patients demonstrated a strong correlation between metastatic recurrence of colon cancer and tumor microvessel density (MVD). The higher the MVD, the more likely tumor cells are to enter the circulatory system, exacerbating the risk of recurrence [13]. Furthermore, Park found that tumor cells originating from larger tumors are more prone to shedding into the abdominal and pelvic cavities, as well as the vascular tissue, further increasing the probability of tumor recurrence post-surgery [14]. Tumor recurrence is comparably prevalent among patients diagnosed with multiple colon cancers. Li [15] assessed this supposition through the implementation of two distinct mouse models. Specifically, mice within the experimental group underwent conventional tumor resection, while mice within the control group underwent sham surgery. Remarkable distinctions were identified in the size of tumor growth and the extent of recurrence within the experimental group compared to the control group.

The findings of the present investigation suggest that postoperative CEA levels may serve as an indicator of the likelihood of colon cancer recurrence in patients. Gold previously regarded CEA as an acidic glycoprotein produced by normal human mucosal cells, which lacked specificity for diagnosing colorectal cancer [16, 17]. However, in recent years, medical testing techniques have advanced and clinicians have come to acknowledge the significance of CEA. An earlier prospective study analyzed the correlation between serum tumor marker concentrations in colon cancer patients and clinical factors, revealing a positive association between elevated CEA levels and colon cancer development [18]. Subsequently, Tsuyoshi et al. reported that most patients experienced a return of their serum CEA concentrations to normal levels three months following radical colon cancer surgery. In contrast, a subset of patients whose postoperative CEA levels did not decrease from preoperative levels had a high risk of rapid tumor recurrence. The elevated CEA levels following surgery can serve as a marker for colon cancer recurrence, which is consistent with the outcomes of the present study [19]. In recent times, some medical practitioners have employed a combination of preoperative CEA, CA19-9, CK-1, and MUC-1 to detect colon cancer in patients diagnosed with the disease. This approach has shown to enhance the sensitivity and specificity of tumor monitoring, as well as assess tumor stage and metastasis more accurately, and is particularly useful in predicting the likelihood of postoperative recurrence in patients [20].

Given that most of the blood flow from the gastrointestinal tract returns via the portal system, the liver is among the most frequently metastasized organs in advanced gastrointestinal tumors, with approximately 20% of colon cancer patients developing liver metastases during the course of their disease [21, 22]. The optimal treatment approach for colon cancer patients with multiple liver metastases involves resection of liver metastases in conjunction with radical colon cancer surgery. However, up to 40% of colon cancer cases remain after surgery, with complete eradication of the tumor proving to be difficult. The present study findings indicate that patients with preoperative liver metastases are at an increased risk of postoperative tumor recurrence. Metastatic colon cancer cells in the liver are known to exist in a dormant state. However, any alteration in the immune system or the organ microenvironment can activate these cells, leading to postoperative recurrence [23]. Liver cells are considered to be stable cells with a high regenerative capacity, but trauma or surgical resection can cause these cells to transition from a stable to a dividing state. Several studies [24–26] have suggested that proliferating liver cells can promote the growth of tumor cells. Residual tumor cells in the liver after surgery may also activate the hepatic epidermal growth factor receptor (EGFR), leading to the promotion of tumor recurrence. Additionally, after hepatectomy, endothelial cell growth factor

(ECGF) is upregulated due to the remodeling of liver vasculature, which can stimulate tumor vascular growth [25, 26]. Hepatocyte growth factor (HGF) is the most potent mitogen that stimulates liver cell proliferation. After hepatectomy, overexpression of HGF also activates dormant residual cancer cells [27–29]. Notably, metastatic liver carcinomas can express matrix metalloproteinase-2 (MMP-2), which is closely associated with tumor recurrence and metastasis. On one hand, MMP-2 can decompose basement membrane glycoproteins and extracellular matrix protein components, thereby promoting tumor invasion and metastasis. Furthermore, MMP-2 can encourage its own secretion by positively regulating MMP-1. On the other hand, MMP-2 also plays a role in promoting tumor vascular proliferation, thereby increasing the risk of tumor recurrence [30, 31].

SHAP analysis has revealed that lymphatic metastasis is a major risk factor for postoperative recurrence in patients with colon cancer. This mechanism is primarily observed in two aspects. Firstly, there exists a dense lymph node network in the colonic mesentery around the tumor, which complicates surgical radical treatment post tumor invasion and limits complete tumor removal. Secondly, tumors frequently metastasize to retroperitoneal organs via lymph node metastasis. Clinical manifestations in patients are often subtle, and imaging examinations pose a challenge in diagnosis. These factors contribute to the elevated risk of postoperative tumor recurrence [32, 33]. David's study [32] similarly found a close correlation between lymph node metastasis and tumor recurrence, and Radespiel's [34] study discovered that a higher number of lymph node metastases lead to an increased chance of tumor recurrence and postoperative mortality rate. Therefore, it is essential for the surgeon to thoroughly clear the pertinent lymph nodes during radical colon cancer surgery, prevent squeezing of the tumor, and avoid tumor dissemination into the abdominal cavity [35].

The present study also examined factors such as surgical approach to evaluate tumor recurrence and found no significant difference between the two approaches, which remains somewhat controversial in clinical practice. Aasmund [36] concluded that laparoscopic surgery adheres to the concept of minimally invasive surgery, which has minimal impact on the patient's immune system and reduces the likelihood of tumor recurrence in postoperative patients. Conversely, Mirow [37] suggested that the trocar used in laparoscopic surgery may cause tumor implantation. Therefore, clinicians should opt for minimally invasive surgical approaches when treating patients with colon cancer to reduce patient trauma. Moreover, operators should strictly adhere to the tumor-free principle and avoid contact with the tumor when inserting the trocar to minimize the risk of tumor dissemination.

In recent years, numerous prediction models have been constructed to predict colon cancer recurrence with varying degrees of success [38–40]. However, many of these models have been constructed using parametric regression which assumes linear relationships between clinical characteristic variables. Unfortunately, patient prognosis cannot be accurately predicted using regression models alone due to the complex interrelationships between clinical variables. To address this, the present study utilized the XGBoost machine learning method to construct a prediction model for tumor recurrence after radical colon cancer surgery that can meet the practical needs of clinical decision making. The proposed model recommends that clinicians utilize a combination of CEA, CA19-9, and other carcinoembryonic antigens for timely follow-up review of postoperative patients. For patients presenting with symptoms such as low back pain or intestinal obstruction, CT and other imaging examinations can also be used to diagnose whether patients have retroperitoneal metastasis. Research conducted by Shibata [32] shows that the survival rate of patients with recurrent colon cancer is low when only radiotherapy and chemotherapy are administered. Resurgical treatment has demonstrated significantly better efficacy than nonsurgical treatment, and surgery remains the primary treatment for patients with recurrent colon cancer [32, 41]. For patients with large tumors or

multiple tumors that cannot be completely resected, chemotherapy should be administered early to reduce tumor size prior to radical resection.

The present study has evaluated the model thoroughly in terms of discrimination, calibration, and clinical utility; however, there are several limitations that should be noted. Firstly, imaging and other related factors were not considered in the study, which might affect the accuracy of the prediction model. The prognosis of tumor patients greatly hinges upon Lynch syndrome, MMR gene, MSI-H, and genetic mutations. However, regrettably, this study lacked the requisite data to conduct a comprehensive predictive analysis in this regard. Nevertheless, we aim to ameliorate this research in the future by gathering pertinent data, thereby offering more advantageous insights for the prognosis of colorectal cancer patients. Additionally, the study was limited to a single center and was conducted retrospectively, which could lead to selection bias, distribution bias, and retrospective bias. Therefore, in future studies, it is recommended to include multicenter prospective studies to increase the reliability and generalizability of the results.

## Conclusion

A model utilizing the XGBoost machine learning algorithm was developed in this study to predict the likelihood of tumor recurrence in colon cancer patients following surgery. The model was found to possess robust predictive accuracy and clinical utility, providing surgeons with an effective diagnostic tool for timely identification of high-risk patients. The model identifies postoperative tumor recurrence as a significant obstacle in the management of CME after surgery, highlighting factors such as postoperative CEA, tumor size, lymphatic and liver metastasis, and number of tumors as closely associated with the risk of recurrence.

## Supporting information

**S1 Checklist. STROBE statement—checklist of items that should be included in reports of observational studies.**
(DOCX)

**S1 Table. Raw data.**
(XLSX)

## Author Contributions

**Conceptualization:** Yuan Liu, Wenyi Du, Wei Shen.

**Data curation:** Yuan Liu, Wenyi Du, Yi Guo.

**Formal analysis:** Yuan Liu, Wenyi Du, Yi Guo, Wei Shen.

**Funding acquisition:** Zhiqiang Tian, Wei Shen.

**Methodology:** Yuan Liu, Wenyi Du, Yi Guo.

**Project administration:** Yuan Liu, Wenyi Du, Yi Guo, Wei Shen.

**Resources:** Yuan Liu, Wenyi Du, Yi Guo, Zhiqiang Tian, Wei Shen.

**Software:** Yuan Liu, Wenyi Du, Yi Guo, Zhiqiang Tian, Wei Shen.

**Supervision:** Zhiqiang Tian, Wei Shen.

**Validation:** Zhiqiang Tian, Wei Shen.

**Visualization:** Zhiqiang Tian, Wei Shen.

**Writing – original draft:** Yuan Liu, Wenyi Du.

**Writing – review & editing:** Zhiqiang Tian, Wei Shen.

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
