## [Decision Letter · Decision Letter 0]

7 Jul 2023

PONE-D-23-08788Identification of High-Risk Factors for Recurrence of Colon Cancer Following Complete Mesocolic Excision: An 8-Year Retrospective StudyPLOS ONE

Dear Dr. Shen,

Thank you for submitting your manuscript to PLOS ONE. After careful consideration, we feel that it has merit but does not fully meet PLOS ONE’s publication criteria as it currently stands. Therefore, we invite you to submit a revised version of the manuscript that addresses the points raised during the review process. Please submit your revised manuscript by Aug 21 2023 11:59PM. If you will need more time than this to complete your revisions, please reply to this message or contact the journal office at plosone@plos.org. Please include the following items when submitting your revised manuscript:A rebuttal letter that responds to each point raised by the academic editor and reviewer(s). You should upload this letter as a separate file labeled 'Response to Reviewers'.A marked-up copy of your manuscript that highlights changes made to the original version. You should upload this as a separate file labeled 'Revised Manuscript with Track Changes'.An unmarked version of your revised paper without tracked changes. You should upload this as a separate file labeled 'Manuscript'.If applicable, we recommend that you deposit your laboratory protocols in protocols.io to enhance the reproducibility of your results. Protocols.io assigns your protocol its own identifier (DOI) so that it can be cited independently in the future. For instructions see: https://journals.plos.org/plosone/s/submission-guidelines#loc-laboratory-protocols. Additionally, PLOS ONE offers an option for publishing peer-reviewed Lab Protocol articles, which describe protocols hosted on protocols.io. Read more information on sharing protocols at https://plos.org/protocols?utm_medium=editorial-email&utm_source=authorletters&utm_campaign=protocols.

We look forward to receiving your revised manuscript.

Kind regards,

Muhammad Tarek Abdel Ghafar, M.D

Academic Editor

PLOS ONE

Journal Requirements:

"This work was supported by the Top Talent Support Program for young and middle-aged people of Wuxi Health Committee (Grant No. HB2020007)."        

Reviewers' comments:

Reviewer's Responses to Questions

**Comments to the Author**

1. Is the manuscript technically sound, and do the data support the conclusions?

Reviewer #1: Yes

Reviewer #2: Yes

2. Has the statistical analysis been performed appropriately and rigorously? 

Reviewer #1: I Don't Know

Reviewer #2: Yes

3. Have the authors made all data underlying the findings in their manuscript fully available?

Reviewer #1: Yes

Reviewer #2: Yes

4. Is the manuscript presented in an intelligible fashion and written in standard English?

Reviewer #1: Yes

Reviewer #2: Yes

5. Review Comments to the Author

Reviewer #1: This is a well written manuscript that addresses the development of a machine learning model that can identify high-risk factors before, during, and after surgery, and to predict postoperative colon cancer recurrence with a high prediction accuracy.

There are some limitation that need to be addressed:

- patients included in this study were operated between 2010 and 2018. Authors should describe why recruitment stopped 5 years ago and it would be valuable to address whether any improvements in surgical techniques occurred during this period. This could significantly modify the results.

-Some references are not recent ones and it would be important to consider updating them, i.e.

11. Benson AB, 3rd, Venook AP, Cederquist L, Chan E, Chen YJ, Cooper HS, et al. Colon Cancer, Version 1.2017, NCCN Clinical Practice Guidelines in Oncology. J Natl Compr Canc Netw. 2017;15(3):370-98. doi: 10.6004/jnccn.2017.0036. The last version available in NCCN.com is V.1.2023 updated in 03/06/23, and there were 3 different versions in 2022.

-I would strongly suggest the model to incorporate genetic/molecular information in a future analysis as a confounding factor. Lynch syndrome is a cancer predisposition syndrome linked to germline pathogenic variants in MMR genes. As one out of 35 colorectal cancers is attributable to Lynch syndrome this information may have an association in the analysis.

Reviewer #2: There are no problems with the strategy, the results, or the method of analysis.

I think it would be difficult to use tumor markers alone, so I would like to see the model evolve to incorporate information on MSI-H and genetic mutations to make it more accurate.

I would like to point out is that the resolution of the diagrams or figures is very poor and needs to be improved.

6. PLOS authors have the option to publish the peer review history of their article (what does this mean?). If published, this will include your full peer review and any attached files.

Reviewer #1: No

Reviewer #2: No

---

## [Author Response · Author response to Decision Letter 0]

12 Jul 2023

Dear reviewers:

Thank you for your decisions and constructive comments on my manuscript. We have carefully considered the suggestion of reviewers and make some changes. We have done our best to improve and made some changes to the manuscript and have marked the changes in red.

Respond to reviewers' and editors' comments:

Journal Requirements:

1.Comment: Please ensure that your manuscript meets PLOS ONE's style requirements, including those for file naming

Response: We have revised the formatting of the manuscript in accordance with the journal's requirements.

2.Comment: Please provide additional details regarding participant consent. In the ethics statement in the Methods and online submission information, please ensure that you have specified (1) whether consent was informed and (2) what type you obtained (for instance, written or verbal, and if verbal, how it was documented and witnessed). If your study included minors, state whether you obtained consent from parents or guardians. If the need for consent was waived by the ethics committee, please include this information. If you are reporting a retrospective study of medical records or archived samples, please ensure that you have discussed whether all data were fully anonymized before you accessed them and/or whether the IRB or ethics committee waived the requirement for informed consent. If patients provided informed written consent to have data from their medical records used in research, please include this information.

Response: Given the retrospective nature of this study and our comprehensive anonymization of all patient data, the Ethics Committee waived the necessity for informed consent for this investigation in adherence to local laws and regulations. This paragraph has been appended to lines 56-58 of the manuscript. We express our gratitude to the esteemed editorial team of the journal for their invaluable suggestions, which have enhanced the rigor of this study.

3.Comment: Thank you for stating the following financial disclosure: "This work was supported by the Top Talent Support Program for young and middle-aged people of Wuxi Health Committee (Grant No. HB2020007)." Please state what role the funders took in the study. If the funders had no role, please state: "The funders had no role in study design, data collection and analysis, decision to publish, or preparation of the manuscript." If this statement is not correct you must amend it as needed. Please include this amended Role of Funder statement in your cover letter; we will change the online submission form on your behalf.

Response: We have added "The funders had no role in study design, data collection and analysis, decision to publish, or preparation of the manuscript." to the Funding section of the manuscript, which can be found on lines 382-384.

4.Comment: Please review your reference list to ensure that it is complete and correct. If you have cited papers that have been retracted, please include the rationale for doing so in the manuscript text, or remove these references and replace them with relevant current references. Any changes to the reference list should be mentioned in the rebuttal letter that accompanies your revised manuscript. If you need to cite a retracted article, indicate the article’s retracted status in the References list and also include a citation and full reference for the retraction notice.

Response: The references have undergone thorough examination, and certain modifications have been implemented in response to the insightful comments provided by the reviewers.

Reviewers' comments:

Reviewer 1#

1.Comment: This is a well written manuscript that addresses the development of a machine learning model that can identify high-risk factors before, during, and after surgery, and to predict postoperative colon cancer recurrence with a high prediction accuracy. There are some limitation that need to be addressed: patients included in this study were operated between 2010 and 2018. Authors should describe why recruitment stopped 5 years ago and it would be valuable to address whether any improvements in surgical techniques occurred during this period. This could significantly modify the results.

Response: We extend our sincere gratitude to the reviewers for their recognition of this study. Simultaneously, we deeply regret any confusion that certain sections of the manuscript may have caused them. The present investigation involved the collection of data from patients who sought medical care at the hospital between 2010 and 2018. Subsequently, these patients were diligently followed for a minimum of 5 years until the conclusion of the current study in 2023. Consequently, patients who visited the hospital in more recent years were not included, as their follow-up duration would have been insufficient to meet the desired study objectives.

2.Comment: Some references are not recent ones and it would be important to consider updating them, i.e. 11. Benson AB, 3rd, Venook AP, Cederquist L, Chan E, Chen YJ, Cooper HS, et al. Colon Cancer, Version 1.2017, NCCN Clinical Practice Guidelines in Oncology. J Natl Compr Canc Netw. 2017;15(3):370-98. doi: 10.6004/jnccn.2017.0036. The last version available in NCCN.com is V.1.2023 updated in 03/06/23, and there were 3 different versions in 2022.

Response: Thanks to the reviewers' suggestions, we have updated the references accordingly.

3.Comment: I would strongly suggest the model to incorporate genetic/molecular information in a future analysis as a confounding factor. Lynch syndrome is a cancer predisposition syndrome linked to germline pathogenic variants in MMR genes. As one out of 35 colorectal cancers is attributable to Lynch syndrome this information may have an association in the analysis.

Response: We wholeheartedly concur with the reviewer's insightful suggestion regarding the pivotal predictive role of genetic molecular information, such as Lynch syndrome and MMR genes, in the prognosis of tumor patients. Regrettably, this study lacks the necessary data to conduct a comprehensive predictive analysis in this regard. As acknowledged in the limitations section of the manuscript (lines 255-258), we have duly highlighted this constraint. Nonetheless, we are committed to collecting relevant data in future endeavors, aiming to continuously enhance this study and provide more beneficial insights for the prognosis of colorectal cancer patients. We thank the reviewers for their suggestions, which have made the article more rigorous.

4.Comment: There are no problems with the strategy, the results, or the method of analysis. I think it would be difficult to use tumor markers alone, so I would like to see the model evolve to incorporate information on MSI-H and genetic mutations to make it more accurate.

Response: Indeed, the prognostication of tumor patients is significantly influenced by genetically inherited molecular information. However, this study lacks the requisite data to conduct a comprehensive predictive analysis in this context, a limitation that has been explicitly acknowledged. As stated in the limitations section of this study, we intend to diligently gather relevant data in subsequent research endeavors, with the aim of continuously refining this study and providing enhanced prognostic assistance for colorectal cancer patients. Thanks to the reviewers for their suggestions, they made the article more rigorous.

5.Comment: I would like to point out is that the resolution of the diagrams or figures is very poor and needs to be improved.

Response: We deeply regret any confusion that may have arisen due to the visual elements, namely the pictures and tables, in the manuscript. We have taken diligent measures to address this issue by processing the images appropriately, resulting in improved clarity and enhanced readability. Thanks to the reviewers' comments, we have further improved the discussion section.

---

## [Decision Letter · Decision Letter 1]

24 Jul 2023

Identification of High-Risk Factors for Recurrence of Colon Cancer Following Complete Mesocolic Excision: An 8-Year Retrospective Study

PONE-D-23-08788R1

Dear Dr. Shen,

We’re pleased to inform you that your manuscript has been judged scientifically suitable for publication and will be formally accepted for publication once it meets all outstanding technical requirements.

Kind regards,

Muhammad Tarek Abdel Ghafar, M.D

Academic Editor

PLOS ONE

Additional Editor Comments (optional):

Reviewers' comments:

Reviewer's Responses to Questions

**Comments to the Author**

1. If the authors have adequately addressed your comments raised in a previous round of review and you feel that this manuscript is now acceptable for publication, you may indicate that here to bypass the “Comments to the Author” section, enter your conflict of interest statement in the “Confidential to Editor” section, and submit your "Accept" recommendation.

Reviewer #2: All comments have been addressed

2. Is the manuscript technically sound, and do the data support the conclusions?

Reviewer #2: Yes

3. Has the statistical analysis been performed appropriately and rigorously? 

Reviewer #2: Yes

4. Have the authors made all data underlying the findings in their manuscript fully available?

Reviewer #2: Yes

5. Is the manuscript presented in an intelligible fashion and written in standard English?

Reviewer #2: Yes

6. Review Comments to the Author

Reviewer #2: Thank you for sending the revised manuscript.　 I believe that all the points raised by the reviewers have been improved.　 This reviewer recommends that this manuscript is acceptable for publication in PLOS One.

7. PLOS authors have the option to publish the peer review history of their article (what does this mean?). If published, this will include your full peer review and any attached files.

Reviewer #2: No

---

## [Editor Report · Acceptance letter]

4 Aug 2023

PONE-D-23-08788R1 

Identification of High-Risk Factors for Recurrence of Colon Cancer Following Complete Mesocolic Excision: An 8-Year Retrospective Study 

Dear Dr. Shen:

I'm pleased to inform you that your manuscript has been deemed suitable for publication in PLOS ONE. Congratulations! Your manuscript is now with our production department. 

Kind regards, 

on behalf of

Prof Muhammad Tarek Abdel Ghafar 

Academic Editor

PLOS ONE